# Achieving Sustainable Tourism: Analysis of the Impact of Environmental Education on Tourists' Responsible Behavior

**Jinming Wang [1,*], Jialu Dai [1,*], Weijun Gao [1], Xingbo Yao [2], Bart Julien Dewancker [1], Jiawei Gao [1], Yuhan Wang [3] and Jiayi Zeng [4]**

1   Faculty of Environmental Engineering, The University of Kitakyushu, Kitakyushu 808-0135, Japan; gaoweijun@me.com (W.G.); bart@kitakyu-u.ac.jp (B.J.D.); jane20001001@gmail.com (J.G.)
2   School of Art and Design, Xi'an University of Technology, Xi'an 710054, China; yaohanhome@gmail.com
3   Department of Architectural Design, The University of Kitakyushu, Kitakyushu 808-0135, Japan; ham309253936@gmail.com
4   Department of Environmental Life Engineering, The University of Kitakyushu, Kitakyushu 808-0135, Japan; zengjiayi0006@gmail.com
*   Correspondence: c1dbb402@eng.kitakyu-u.ac.jp (J.W.); c1dbb410@eng.kitakyu-u.ac.jp (J.D.)

**Abstract:** Environmental challenges like resource depletion, climate change, and biodiversity loss require human action. Environmental education influences individuals' understanding and motivates responsible behavior, making it a real solution to address current problems. In this study, we examine the impact of situational environmental education and daily environmental education on tourists' responsible environmental behavior by incorporating them as covariates into an integrated tourist behavior model. In total, 2381 valid questionnaires were obtained. The results showed that (1) daily environmental education mainly affects tourists' responsible environmental behavior through attitude (0.467) and habits (0.634); (2) tourists' responsible environmental behavior is mainly affected by situational environmental education through habits (0.534), subjective norms (0.504), and intention (0.614); and (3) personal factors, including attitudes toward environmental behavior, subjective norms, and perceptual behavior control, positively influence tourists' responsible environmental behavioral intention and their responsible environmental behavior. This research will help to improve sustainability indicators and frameworks and to promote the adoption of sustainable tourism practices.

**Keywords:** behavioral interventions; environmental education; sustainability indicators and frameworks; sustainable tourism; tourists' responsible environmental behavior

## 1. Introduction

### 1.1. Background

Humanity is currently confronted with a range of critical environmental challenges that cannot be ignored, including climate change, natural resource depletion, and biodiversity loss [1]. To meet these challenges, an international and interdisciplinary human effort is necessary [2]. Researchers generally believe that current human behavior has a negative impact on the earth's environment [3]. The rapid growth in tourism in recent years has put tourist destinations under enormous pressure [4]. Most tourists have very poor self-control and environmental awareness, which leads to unwanted behavior [5–7]. These actions have caused serious ecological problems [8], which have become even more evident during the COVID-19 pandemic [9–11]. If human interventions in the environment are reduced, nature has the potential to restore itself [12]. The COVID-19 pandemic had a short-term positive impact on the air quality in China and played a notable role in reducing global carbon emissions [10]. Relevant studies have also found that emergency policies adopted due to the epidemic had a significant impact on beach cleanliness and environmental noise [9]. These

findings emphasize the necessity of improving environmental behavior; among methods to achieve this, environmental education is a key tool in promoting sustainable tourism.

### 1.2. The Role of Environmental Education

Environmental education is a powerful means of addressing environmental challenges as it seeks to promote the values and practices of environmental preservation and sustainability [13,14]. Environmental education goes beyond simply imparting knowledge and understanding the natural world. It also aims to inspire individuals to take action and make informed decisions that promote environmental sustainability [15]. The promotion of sustainable development requires a strong emphasis on education as a fundamental component. Education is vital in raising awareness and understanding of the complicated environmental issues we need to face and in equipping individuals with the abilities and knowledge needed to solve these challenges [16]. To effectively mitigate the undesirable influence of human behavior on the environment, it is important to comprehend the underlying intrinsic motivations that drive human behavior [17]. Extrinsic motivations may produce only temporary changes in behavior [18]. Therefore, nurturing intrinsic motivations for responsible environmental behavior is essential for achieving significant and lasting reductions in human-induced environmental problems. Many scholars believe that environmental education, besides offering suitable information, can also promote intrinsic motivation [19–21]. Most of the present research tends to concentrate on the environmental education that people receive throughout their childhood [16,22], neglecting the environmental education people receive in tourist destinations and the mechanisms of its influence on responsible environmental behavior. This unconscious environmental education provided by tourist destinations often has an important influence on tourists' responsible environmental behavior [23]. Therefore, it is vital to research education's effect on tourists' responsible environmental behavior and identify the most effective approaches for intervention.

### 1.3. Legal Frameworks and Educational Strategies in Environmental Protection

In addition to the observed environmental impacts of human behavior and global challenges, it is critical to recognize emerging legal frameworks and educational approaches aimed at protecting the natural world. Notably, academics such as Panigaj and Berníková have discussed the prospect of "ecocide" as a potential international crime, elucidating its legal implications and identifying the need for international cooperation [24]. Similarly, Cristina Aragão Seia explored the nuances of environmental liability and the necessity of amending European legislation to better address environmental issues [25]. Furthermore, the study by Walter D. Gaveni and Kola O. Odeku provides a detailed analysis of international legal instruments related to environmental duty of care, emphasizing the need to hold perpetrators responsible for environmental damage [26]. Finally, the study by Majerčáková and Mittelman highlights the importance of waste behavior within the broader area of environmental protection, illustrating the complex relationships between legislative measures and environmental sustainability [27]. Integrating insights from these studies into our research not only enriches the theoretical framework but also better promotes sustainable tourism development from the overall perspective of environmental education and legal frameworks.

### 1.4. Environmental Knowledge and Situational Factors

Acquiring environmental knowledge is regarded as an essential condition for engaging in ecological behavior [28,29]. While engagement in environmental education initiatives typically leads to positive effects on an individual's understanding and awareness of environmental issues [30], the relationships between behavior and environmental knowledge are debated [28,31], and environmental education outcomes can be influenced by various factors, including personal attitudes and values [32]. In fact, in tourist destinations, besides positively influencing environmental knowledge, the environmental education received

by tourists will, as a situational factor, further influence tourists' motivation to behave environmentally responsibly. For example, the environmental lessons received by some tourists in scenic areas do not positively influence their environmental knowledge, but they still develop positive and responsible environmental behaviors. Environmental education provided by a tourist destination can also have an impact on tourists' attitudes, habits, and many other aspects.

Situational factors are the total factors that people directly associate with the perceiver in the perception process. Situational factors pertain to "the sum of all those factors that are specific to the time and place of observation in the process of people's perception" [33]. Research in this area has shown that situational factors can have an important influence on tourists' responsible environmental behavior [6,7]. For tourists' perception, situational factors' influence on human perception varies based on the specific characteristics of these factors but more significantly on the connection between situational factors and the perceiver and the perceived. During a tour, tourists are in a state of being perceived, and the environmental education of the tourist destination is directly related to the tourist. Environmental education in tourist destinations can be used as a situational factor embedded in the tourist comprehensive analysis model for analysis and verification.

### 1.5. Research Purpose and Significance

This study is based on a comprehensive theoretical framework based on environmental psychology and pedagogy. Through theoretical assumptions and data analysis, it reveals the mechanisms by which environmental education influences tourists' responsible behaviors, thereby providing a scientific basis for the sustainable development of tourist destinations. In previous research, environmental education has often been treated as a covariate influencing attitudes [34,35]. This study presents a novel perspective and contribution by addressing the gap in existing research regarding the mechanisms by which different types of environmental education impact responsible environmental behavior. The primary goal is to reveal the mechanisms by which environmental education influences tourists' responsible behaviors. The second goal is to compare the effectiveness of situational education and daily environmental education in this context, delving into how both situational and daily environmental education shape tourists' attitudes and behaviors towards the environment. This research is crucial in the context of sustainable tourism, where effective strategies to mitigate the negative impacts of tourism on the environment are urgently required. It contributes to a broader discussion on sustainability and informs the practices of policymakers, educators, and the tourism industry.

## 2. Theoretical Basis

### 2.1. Theory of Planned Behavior (TPB)

Ajzen's theory of planned behavior is presented as a common model of deliberate action [36]. The core assumption is that the intention to perform a specific behavior—in other words, the resolve to seek justification for that behavior—directly influences the behavior itself. The likelihood of a person behaving a certain way is affected by their attitude towards that behavior, the subjective norms surrounding that behavior, and their perceived ability to control or carry out the behavior [36]. According to this theory, behavioral intention directly influences behavior. Intentions are formed by evaluating three different factors in the rational choice process: the attitude toward behavior (ATEB); subjective norms (SNs), which are an individual's perception of the social pressure to behave in a specific manner and the personal perception of the control of behavior in a condition; and perceived behavioral control (PBC), which is the experience of a person in full command of the condition or at the very least partially commanded by other people or the condition of the situation. In the behavioral field, the TPB has been proven to be helpful in explaining the choice of travel mode [37], recycling behavior [38], water conservation [39], and ecological consumer behavior [7].

### 2.2. Norm Activation Theory (NAT) and Value–Belief–Norm Theory (VBN)

While the TPB is a comprehensive theory that can be applied to a huge range of behaviors, the Norm Activation Theory (NAT) was initially created to describe the specific behaviors of altruism and helping behavior. In other words, while TPB provides a general framework for understanding behavior, NAT is more focused and tailored to a specific type of behavior [2,40]. This theory assumes that behavior is driven by a sense of moral obligation (personal norm). Such personal norms are not automatically triggered but need to be activated through a specific process. In other words, this process begins only when a person feels a need for someone or something (awareness of need). An individual will only take action if they perceive a cause-and-effect relationship between their behavior and the intended outcome (awareness of consequences). Actors must experience a certain level of perceived behavioral control in order to activate personal norms. Although Norm Activation Theory (NAT) was originally created to explain altruistic behavior, its applicability to behavior related to specific contexts is not immediately obvious. However, Thøgersen's perspective suggests that environmental behavior is influenced by ethical considerations and personal values, in addition to practical considerations, such as weighing the costs and benefits of certain actions [41]. Following Thøgersen, many scholars have utilized NAT to understand environment-related behavior. Research has shown that factors related to personal norms, moral obligations, and emotional responses—which are key components of NAT—can play a role in influencing behaviors related to protecting the environment [42,43]. While TPB considers both moral and non-moral motivations that drive environmental behavior, NAT places more emphasis on moral factors and tends to overlook the non-moral drivers that TPB would account for.

Stern's Value–Belief–Norm Theory (2000) aims to connect the principles of NAT with research findings on how overall values, environmental convictions, and actions are interrelated. Therefore, it is also a comprehensive theory in itself. It assumes that behavior is directly determined by NAT-based personal norms. Stern's theory suggests that NAT-based personal norms influence an individual's behavior, and he proposed that personal norms can be triggered through the awareness of consequences and attribution of responsibility. Additionally, he suggested that there is a causal relationship between these two factors, with awareness of consequences being a crucial prerequisite for attribution of responsibility. VBN theory has been used in environmental studies and has empirical support [43–45].

### 2.3. Environmental Education

Environmental education is an educational method designed to integrate environmental concerns at the very core of the educational process, aiming to foster environmental sustainability [46]. It aims to develop environmentally literate citizens who are well suited to address environmental and resource sustainability problems [47]. Environmental education encourages individuals to cultivate the necessary attitudes, values, knowledge, dispositions, and skills for engaging in environmental action, thereby enhancing their involvement in improving the long-term viability of the relationships between humans and the natural environment [48,49]. The model of situational environmental education is centered on the principles of preservation and sustainability and is designed to align with natural processes and systems [50]. Objective situational environmental knowledge education involves providing tourists with information about the environment at a destination, with a focus on the type and nature of the information being conveyed [28,51]. Environmental-education-themed destinations can offer Citizen Science (CS), education for sustainable development (ESD), experiential education (ExE), garden-based learning (GBL), inquiry-based education (IBE), and other educational models. These educational models can interact with individual tourist factors [52]. Several studies have shown that environmental interpretations in environmental education, including management signs ("no smoking", etc.), educational signs (policies, plans, landscape guides, etc.), reward and punishment signs (environmental incentives), and persuasive signs ("Please don't step on the grass", etc.), positively affect tourists' responsible environmental behavior.

## 3. Research Methodology

### 3.1. Study Region

We selected the Changchun Water Culture Ecological Park as the site for our situational environmental education research project (Figure 1a,b). This park is an important urban development project located in Changchun City. The scenic spot uses animated water purification demonstrations (Figure 1c), water purification equipment, information about water environmental protection, etc., so that tourists can experience (ExE) scientific education related to water purification (IBE), water supply, and drainage processes and water environmental protection and governance. The design of the Changchun Water Culture Ecological Park emphasizes a systematic approach to environmental protection, featuring several different protection systems (Figure 1d), such as a slow-moving system, preserving the natural ecosystem and implementing a self-purifying water ecology system. Forest corridors that pass through the park serve to reduce the impact on native vegetation systems and preserve it as much as possible. At the same time, the forest corridor also provides visitors with a pleasant and invigorating walking experience; it also disseminates environmental education (OE) through various information boards and water purification handicraft displays. The preserved clean water tanks and concrete sedimentation tanks not only enhance the scenic features and minimize damage to the environment but also allow tourists to experience water culture education (GBL) to the maximum. The project is designed to integrate environmental education into the tourist experience, using green spaces as a platform to deliver this education in a way that is engaging and enjoyable for tourists.

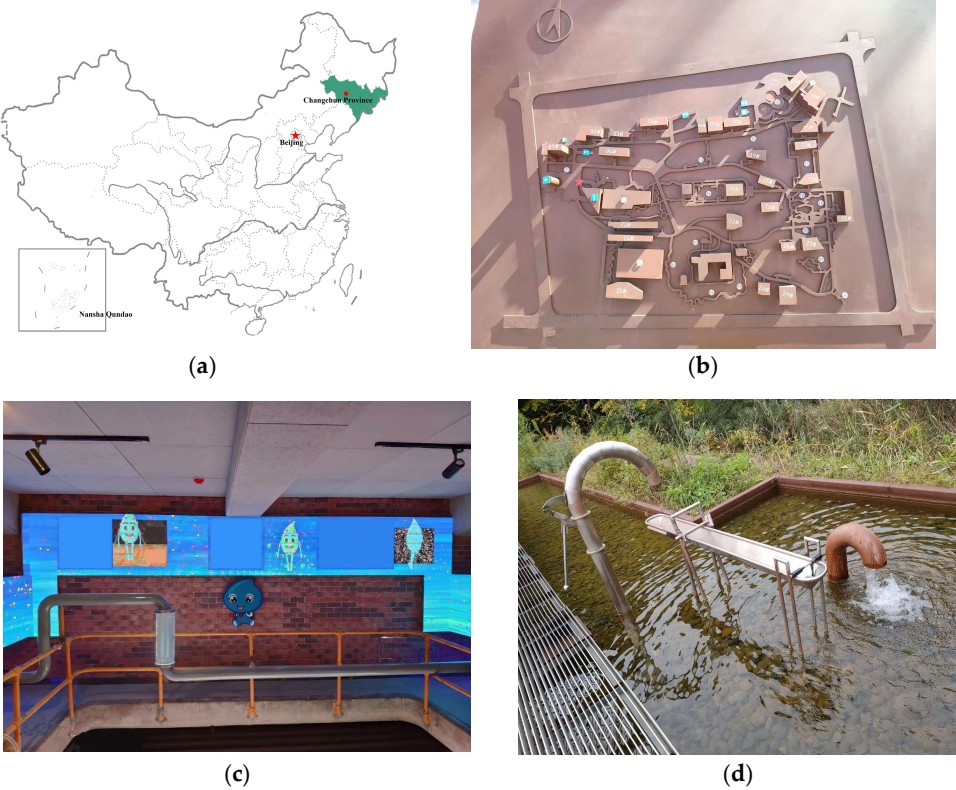

(**a**)          (**b**)

(**c**)          (**d**)

**Figure 1.** Photos and maps of the study region: (**a**) location of Changchun; (**b**) aerial view of Water Culture Ecological Park; (**c**) water purification flowchart; (**d**) water purification plant sites.

We chose to conduct the daily environmental education survey online. The questionnaires were collected via a snowball model as well as a paid completion model.

### 3.2. Questionnaire Design

Consulting previous studies, we developed a questionnaire (Table 1) to examine the factors that influence tourists' responsible environmental behavior. In the model used in this study, tourists' responsible environmental behavior is directly affected by intentions. Attitudes, subjective norms, perceived behavioral control, habits, awareness of consequences, self-transcendence values, self-enhancement values, and environmental education are used as covariates that jointly affect the generation of intention and the occurrence of behaviors. Based on the research model, corresponding hypotheses are proposed. The influencing factors are all measured using a five-point Likert scale [53].

**Table 1.** Scales and hypotheses used in the study.

| Latent Variable | Observed Variable | Item Text | References |
|---|---|---|---|
| ATEB | ATEB1 | In my opinion, it is sensible to preserve the environment of the scenic spot. | |
| | ATEB2 | In my opinion, it is necessary to preserve the tourist attraction environment. | [36,54–56] |
| | ATEB3 | In my opinion, it is valuable to preserve the environment of the scenic spot. | |
| | ATEB4 | In my opinion, it is worth advocating to preserve the tourist attraction environment. | |
| | | **H1:** Tourists' ATEB affects their REBI positively. | |
| SN | SN1 | People who are important in my life believe that it is crucial to preserve the environment of the scenic spot. | |
| | SN2 | People who are important in my life will blame me for not preserving the tourist attraction environment. | [7,23,36] |
| | SN3 | Individuals who hold significance in my life are preserving the tourist attraction environment. | |
| | | **H2:** Tourists' SN affects their REBI positively. | |
| PBC | PBC1 | In my opinion, I can preserve the tourist attraction environment. | |
| | PBC2 | Preserving the tourist attraction environment is an easy task for me. | [36,54–56] |
| | PBC3 | Preserving the tourist attraction environment is a positive thing for me. | |
| | | **H3:** Tourists' PBC affects their REBI positively. | |
| | | **H4:** Tourists' PBC affects their REB positively. | |
| REBI | REBI1 | I want to prevent the destruction of the beautiful natural landscape. | |
| | REBI2 | I want to dispose of the waste produced while traveling in a responsible manner. | [7,23,36] |
| | REBI3 | I prefer not to purchase products that have a negative impact on the ecology of the scenic area. | |
| | | **H5:** Tourists' REBI affects their REB positively. | |
| REB | REB1 | I will safeguard the natural beauty of the tourist attraction from any harm. | |
| | REB2 | I will ensure proper disposal of any waste generated during my travels. | [7,23,36] |
| | REB3 | I will refrain from purchasing products that harm the tourist attraction environment. | |
| AC | AC1 | The punishment for damaging the environment in the scenic area will make me want to preserve the scenic environment. | |
| | AC2 | I would like to preserve the scenic environment because of the national customized laws and regulations. | [2] |
| | AC3 | I would like to preserve the scenic environment because of the effect on my conscience. | |
| | | **H6:** Tourists' AC affects their SN positively. | |
| ST | ST1 | Human beings should be bound by the laws of nature. | |
| | ST2 | Our damage to the environment is approaching the limit that the earth can bear. | [2] |
| | ST3 | Human beings lack the entitlement to alter the natural environment to fulfill their requirements. | |
| | ST4 | Human beings are significantly mistreating the environment. | |
| | | **H7:** Tourists' ST affect their SN positively. | |
| SE | SE1 | Human capacity can ensure that the earth is always habitable. | |
| | SE2 | The natural capacity is sufficient to cope with human industrial development. | |
| | | **H8:** Tourists' SE affects their SN positively. | |

**Table 1.** *Cont.*

| Latent Variable | Observed Variable | Item Text | References |
|---|---|---|---|
| HAB | HAB1 | I have been doing my best to preserve the environment. | [57,58] |
| | HAB2 | I am willing to protect the environment all the time. | |
| | | **H9:** Tourists' HAB affects their REB positively. | |
| sEE | sEE1 | The publicity around environmental protection in the scenic area will help me protect the environment. | [23] |
| | sEE2 | The beautiful scenery in the scenic area makes me want to protect the environment. | |
| | sEE3 | The publicity of cultural relic protection in the scenic area makes me want to protect the environment more. | |
| | | **H10a:** sEE affects tourists' ATEB positively. | |
| | | **H11a:** sEE affects tourists' SN positively | |
| | | **H12a:** sEE affects tourists' REBI positively. | |
| | | **H13a:** sEE affects tourists' REB positively. | |
| | | **H14a:** sEE affects tourists' HAB positively. | |
| dEE | dEE1 | I can often learn about environmental protection in my daily life. | |
| | dEE2 | In daily life, I will actively learn about environmental protection. | |
| | dEE3 | The publicity around cultural relic protection in daily life makes me want to protect the environment more. | |
| | dEE4 | The publicity around natural environment protection in daily life makes me want to protect the environment more. | |
| | | **H10b:** dEE affects tourists' ATEB positively. | |
| | | **H11b:** dEE affects tourists' SN positively. | |
| | | **H12b:** dEE affects tourists' REBI positively. | |
| | | **H14b:** dEE affects tourists' HAB positively. | |

Subjective norm (SN); perceptual behavior control (PBC); responsible environmental behavior (REB); responsible environmental behavioral intention (REBI); attitude toward environmental behavior (ATEB); habit (HAB); awareness of consequences (ACs); self-transcendence values (STs); self-enhancement values (SEs); daily environmental education (dEE); situational environment education (sEE).

The study adhered to the principles outlined in the Declaration of Helsinki [54] and all actions were in accordance with these principles. Participation in the study was voluntary and all individuals who took part provided their consent for their data to be used. The purpose of the experiment was clearly communicated to all participants through the questionnaires, which were completed anonymously to protect their privacy.

*3.3. Data Acquisition*

The survey was carried out from 16 September 2020 to 14 January 2022. The questionnaires were conducted both online and offline. A total of 1529 questionnaires were collected offline; a total of 16 invalid questionnaires were eliminated according to the lying coefficient, and thus 1513 valid questionnaires were collected. The effective rate of offline questionnaires was 98.94%. A total of 891 questionnaires were collected online. By eliminating 42 invalid questionnaires via the lying coefficient, in total, 868 questionnaires were obtained and considered valid. The effective rate of the online questionnaire was 97.4%. The total number of valid questionnaires was 2381. The data were entered into SPSS 22.0 and underwent a three-step validation process to ensure their accuracy. First, one person entered the data. Then, a second person confirmed the data. Finally, a third person performed a sample check to ensure the accuracy of the data.

*3.4. Analysis Processing*

The hypothesized relationships between the variables in the environmental education comprehensive model were tested using structural equation modeling (SEM) statistical methods. To evaluate the internal structure, reliability, and validity of the comprehensive model, we employed a confirmatory factor analysis (CFA) in AMOS 21.

## 4. Results

### 4.1. Descriptive Statistical Analysis of the Sample

We analyzed the tourists who participated in situational environmental education. Table 2 displays the characteristics of the 753 surveyed tourists, which are described as follows.

(1)  The number of tourists over 60 years old was relatively small, and there were more young and middle-aged visitors as well as visitors with children. The reasons for this may be related to the park's theme.
(2)  The educational level of tourists was generally high, with respondents being mainly college students. The reason for this is that Changchun is an industrial city and most of the colleges and universities are mainly engineering institutions. The theme of the park is also most popular among engineering college students.
(3)  The main audience of environmental education theme parks is students or parents with children. Systematic tours require a certain level of education.
(4)  The vast majority of tourists believe that scenic spots regularly provide them with knowledge about environmental protection. This means that environmental education in tourist destinations has attracted the attention and interest of tourists.

**Table 2.** Demographic data of tourists at Changchun Water Culture Ecological Park.

| Characteristics | Category | Quantity | Percent (%) |
|---|---|---|---|
| Gender | Male | 387 | 51.4 |
| | Female | 366 | 48.6 |
| | ≤10 | 7 | 0.93 |
| | 11–20 | 47 | 6.2 |
| | 21–30 | 277 | 36.8 |
| Age | 31–40 | 167 | 22.2 |
| | 41–50 | 156 | 20.7 |
| | 51–60 | 86 | 11.4 |
| | ≥60 | 13 | 1.7 |
| | Elementary school and below | 16 | 2.1 |
| Education | High school | 295 | 39.2 |
| | College | 367 | 48.7 |
| | Masters degree or above | 75 | 9.9 |
| | Public official | 46 | 6.1 |
| | Business personnel | 34 | 4.5 |
| | Mechanics/workers | 159 | 21.1 |
| Occupation | Waiters/salespersons | 27 | 3.6 |
| | Company staff | 196 | 26.0 |
| | Student | 129 | 17.1 |
| | Retired | 64 | 8.5 |
| | Others | 98 | 13.0 |
| | Often | 450 | 59.8 |
| I can learn about | Usually | 237 | 31.5 |
| environmental protection in | Occasionally | 55 | 7.3 |
| the park. | Rarely | 7 | 0.9 |
| | Never | 4 | 0.5 |

### 4.2. Confirmatory Factor Analysis

Comprehensive reliability (CR) is used to assess the reliability of different inner connections. Cronbach's Alpha was utilized to assess the validity, reliability, and internal consistency of the questionnaire. Table 3 displays the $\alpha$ values of the measurement scales, which range from 0.706 to 0.887, which is above the standard of 0.7, indicating good reliability and internal consistency. Thus, the dataset itself demonstrates good internal reliability and consistency [59,60]. The comprehensive reliability values of the six latent

variables ranged from 0.867 to 0.930, higher than the standard of 0.6, indicating a high consistency [61].

**Table 3.** The results of reliability and convergent validity.

| Variable | Mean | SD | Standardized Factor Loading | CR | AVE | Cronbach's Alpha |
|---|---|---|---|---|---|---|
| ATEB | | | | 0.921 | 0.746 | 0.881 |
| ATEB1 | 4.67 | 0.528 | 0.899 | | | |
| ATEB2 | 4.63 | 0.554 | 0.902 | | | |
| ATEB3 | 4.66 | 0.519 | 0.911 | | | |
| ATEB4 | 4.56 | 0.564 | 0.728 | | | |
| SN | | | | 0.871 | 0.692 | 0.706 |
| SN1 | 4.22 | 0.827 | 0.810 | | | |
| SN2 | 3.88 | 0.888 | 0.844 | | | |
| SN3 | 4.08 | 0.734 | 0.841 | | | |
| PBC | | | | 0.893 | 0.736 | 0.750 |
| PBC1 | 4.06 | 0.815 | 0.871 | | | |
| PBC2 | 4.17 | 0.758 | 0.808 | | | |
| PBC3 | 4.26 | 0.619 | 0.892 | | | |
| REBI | | | | 0.906 | 0.763 | 0.841 |
| REBI1 | 4.29 | 0.590 | 0.896 | | | |
| REBI2 | 4.36 | 0.636 | 0.848 | | | |
| REBI3 | 4.25 | 0.635 | 0.875 | | | |
| REB | | | | 0.921 | 0.794 | 0.848 |
| REB1 | 4.38 | 0.635 | 0.899 | | | |
| REB2 | 4.45 | 0.594 | 0.895 | | | |
| REB3 | 4.36 | 0.675 | 0.879 | | | |
| sEE | | | | 0.930 | 0.815 | 0.887 |
| sEE1 | 4.44 | 0.621 | 0.897 | | | |
| sEE2 | 4.50 | 0.578 | 0.883 | | | |
| sEE3 | 4.50 | 0.632 | 0.928 | | | |
| dEE | | | | 0.921 | 0.745 | 0.881 |
| dEE1 | 4.03 | 0.802 | 0.846 | | | |
| dEE2 | 4.00 | 0.784 | 0.879 | | | |
| dEE3 | 4.19 | 0.670 | 0.873 | | | |
| dEE4 | 4.25 | 0.613 | 0.853 | | | |
| AC | | | | 0.867 | 0.685 | 0.766 |
| AC1 | 4.11 | 0.799 | 0.819 | | | |
| AC2 | 4.07 | 0.857 | 0.887 | | | |
| AC3 | 4.13 | 0.853 | 0.773 | | | |
| ST | | | | 0.868 | 0.622 | 0.776 |
| ST1 | 4.13 | 0.801 | 0.816 | | | |
| ST2 | 4.18 | 0.821 | 0.782 | | | |
| ST3 | 4.12 | 0.860 | 0.777 | | | |
| ST4 | 4.12 | 0.814 | 0.778 | | | |
| SE | | | | 0.869 | 0.768 | 0.727 |
| SE1 | 3.03 | 1.150 | 0.888 | | | |
| SE2 | 3.07 | 1.164 | 0.864 | | | |
| HAB | | | | 0.932 | 0.872 | 0.863 |
| HAB1 | 4.30 | 0.600 | 0.938 | | | |
| HAB2 | 4.43 | 0.574 | 0.929 | | | |

We tested the effectiveness of measuring model convergence with standardized factor loadings and the average variance extracted (AVE) (Table 4). The standard factor loadings of the 34 observed variables ranged from 0.728 to 0.938, exceeding the standard of 0.5, suggesting that each observed variable possesses strong explanatory power for its corre-

sponding latent variable [62]. All latent variables' AVE values fell between 0.622 and 0.872, exceeding the standard of 0.5, suggesting that the questionnaire items' average explanatory power was adequate [61]. To test discriminant validity, we compared the correlation coefficient between the latent variables with the square root of their respective means.

**Table 4.** Daily environmental education correlation analysis.

| Latent Variable | ATEB | SN | PBC | REBI | REB | dEE | AC | ST | SE | HAB |
|---|---|---|---|---|---|---|---|---|---|---|
| ATEB | 0.864 | | | | | | | | | |
| SN | 0.398 ** | 0.832 | | | | | | | | |
| PBC | 0.469 ** | 0.626 ** | 0.858 | | | | | | | |
| REBI | 0.612 ** | 0.444 ** | 0.606 ** | 0.874 | | | | | | |
| REB | 0.589 ** | 0.458 ** | 0.606 ** | 0.859 ** | 0.891 | | | | | |
| dEE | 0.467 ** | 0.504 ** | 0.635 ** | 0.663 ** | 0.671 ** | 0.864 | | | | |
| AC | 0.370 ** | 0.463 ** | 0.490 ** | 0.545 ** | 0.552 ** | 0.545 ** | 0.828 | | | |
| ST | 0.408 ** | 0.423 ** | 0.498 ** | 0.560 ** | 0.582 ** | 0.503 ** | 0.538 ** | 0.789 | | |
| SE | 0.017 | 0.117 ** | 0.200 ** | 0.104 ** | 0.131 ** | 0.343 ** | 0.220 ** | 0.075 | 0.877 | |
| HAB | 0.447 ** | 0.445 ** | 0.617 ** | 0.629 ** | 0.641 ** | 0.634 ** | 0.506 ** | 0.536 ** | 0.193 ** | 0.929 |

Note: ** indicates that the significance level is below 0.01 (or 99% confidence level).

The size of the sample was 2381 (over 50); thus, we applied the Jarque–Bera test. According to Table 5, the kurtosis had an absolute value below 10, the skewness had an absolute value below 3, and all $p$-values exceeded 0.005. Thus, we believe that ATEB, SN, PBC, EE, REBI, AC, ST, SE, HAB, and REB all have normal distributions.

**Table 5.** Analysis results of a normality test.

| Latent Variable | Kurtosis | Skewness | $\chi^2$ | $p$ |
|---|---|---|---|---|
| ATEB | −0.204 | 0.190 | 3.577 | 0.167 |
| SN | 0.265 | −0.561 | 2.603 | 0.272 |
| PBC | −0.085 | −0.360 | 0.789 | 0.674 |
| REBI | −0.475 | 0.393 | 4.063 | 0.131 |
| REB | −0.157 | −0.990 | 4.570 | 0.102 |
| dEE | 0.158 | −0.834 | 3.428 | 0.180 |
| AC | 0.064 | −0.942 | 3.863 | 0.145 |
| ST | −0.259 | −0.195 | 1.331 | 0.514 |
| SE | −0.333 | −0.467 | 2.851 | 0.240 |
| HAB | −0.200 | −0.403 | 1.463 | 0.481 |

Table 6 illustrates that the correlation coefficients with other latent variables are exceeded by the square roots of all latent variables, showing discriminant validity. The Pearson correlation coefficient is less than 0.01, so the variables exhibit a good fit in linear regression, and a strong correlation is observed between them.

**Table 6.** Situational environment education correlation analysis.

| Latent Variable | ATEB | SN | PBC | REBI | REB | sEE | AC | ST | SE | HAB |
|---|---|---|---|---|---|---|---|---|---|---|
| ATEB | 0.864 | | | | | | | | | |
| SN | 0.465 ** | 0.832 | | | | | | | | |
| PBC | 0.416 ** | 0.620 ** | 0.858 | | | | | | | |
| REBI | 0.515 ** | 0.549 ** | 0.630 ** | 0.874 | | | | | | |
| REB | 0.475 ** | 0.553 ** | 0.593 ** | 0.869 ** | 0.891 | | | | | |
| sEE | 0.462 ** | 0.504 ** | 0.564 ** | 0.812 ** | 0.804 ** | 0.903 | | | | |
| AC | 0.228 ** | 0.416 ** | 0.473 ** | 0.535 ** | 0.510 ** | 0.531 ** | 0.828 | | | |
| ST | 0.266 ** | 0.397 ** | 0.507 ** | 0.659 ** | 0.630 ** | 0.676 ** | 0.587 ** | 0.789 | | |
| SE | 0.030 | 0.117 | 0.077 | 0.139 * | 0.158 * | 0.135 * | 0.201 ** | 0.157 ** | 0.877 | |
| HAB | 0.433 ** | 0.526 ** | 0.617 ** | 0.764 ** | 0.768 ** | 0.812 ** | 0.550 ** | 0.665 ** | 0.184 ** | 0.929 |

Note: * indicates that the significance level is below 0.05 (or 95% confidence level). ** indicates that the significance level is below 0.01 (or 99% confidence level).

### 4.3. Structural Equation Model

The model fitting results are illustrated in Tables 7 and 8 and Figure 2.

**Table 7.** Path analysis of the structural model (dEE).

| Path | Standard Error | Standardized Path Coefficient | Hypothesis |
|------|----------------|-------------------------------|------------|
| ATEB → REBI | 0.037 | 0.415 | Verified |
| SN → REBI | 0.025 | 0.238 | Verified |
| PBC → REBI | 0.033 | 0.390 | Verified |
| dEE → REBI | 0.030 | 0.373 | Verified |
| REBI → REB | 0.027 | 0.715 | Verified |
| PBC → REB | 0.023 | 0.095 | Verified |
| HAB → REB | 0.026 | 0.137 | Verified |
| dEE → HAB | 0.027 | 0.634 | Verified |
| PBC → HAB | 0.038 | 0.143 | Verified |
| REBI → HAB | 0.040 | 0.312 | Verified |
| SN → HAB | 0.028 | 0.064 | Verified |
| dEE → SN | 0.049 | 0.333 | Verified |
| AC → SN | 0.045 | 0.217 | Verified |
| ST → SN | 0.045 | 0.142 | Verified |
| SE → SN | 0.025 | 0.056 | Verified |
| dEE → ATEB | 0.026 | 0.467 | Verified |

**Table 8.** Path analysis of the structural model (sEE).

| Path | Standard Error | Standardized Path Coefficient | Hypothesis |
|------|----------------|-------------------------------|------------|
| ATEB → REBI | 0.074 | 0.331 | Verified |
| SN → REBI | 0.050 | 0.395 | Verified |
| PBC → REBI | 0.057 | 0.417 | Verified |
| sEE → REBI | 0.045 | 0.614 | Verified |
| sEE → REB | 0.063 | 0.202 | Verified |
| REBI → REB | 0.062 | 0.574 | Verified |
| PBC → REB | 0.040 | 0.023 | Verified |
| HAB → REB | 0.060 | 0.151 | Verified |
| sEE → HAB | 0.063 | 0.529 | Verified |
| PBC → HAB | 0.049 | 0.159 | Verified |
| REBI → HAB | 0.070 | 0.209 | Verified |
| SN → HAB | 0.042 | 0.046 | Verified |
| sEE → SN | 0.067 | 0.504 | Verified |
| AC → SN | 0.067 | 0.275 | Verified |
| ST → SN | 0.076 | 0.232 | Verified |
| SE → SN | 0.035 | 0.026 | Verified |
| sEE → ATEB | 0.047 | 0.462 | Verified |

(1) The normalized path coefficients from ATEB to REBI are 0.331 and 0.415, indicating a high positive correlation between ATEB and REBI. The normalized path coefficients from SN to REBI are 0.395 and 0.238, suggesting that SN is highly positively correlated with REBI. The normalized path coefficients from PBC to REBI are 0.417 and 0.390, and the normalized path coefficients to REB are 0.023 and 0.095, indicating a high correlation between PBC, REBI, and REB. The normalized path coefficients from REBI to REB are 0.574 and 0.715; therefore, there is a strong correlation between PBC and REBI as well as REB. Hypotheses H1, H2, H3, H4, and H5 are all verified.

(2) The normalized path coefficient from AC to SN is between 0.217 and 0.275; therefore, AC and SN are highly positively correlated. The normalized path coefficient from ST to SN is between 0.142 and 0.232, indicating that ST is highly positively correlated with SN. The normalized path coefficient from SE to SN is between 0.056 and 0.026, indicating a high correlation between SE and SN. Hypotheses H6, H7, and H8 are all verified.

(3)   The normalized path coefficient from sEE to REBI is 0.614, suggesting that sEE is highly positively correlated with REBI. Hypothesis H10a is verified.

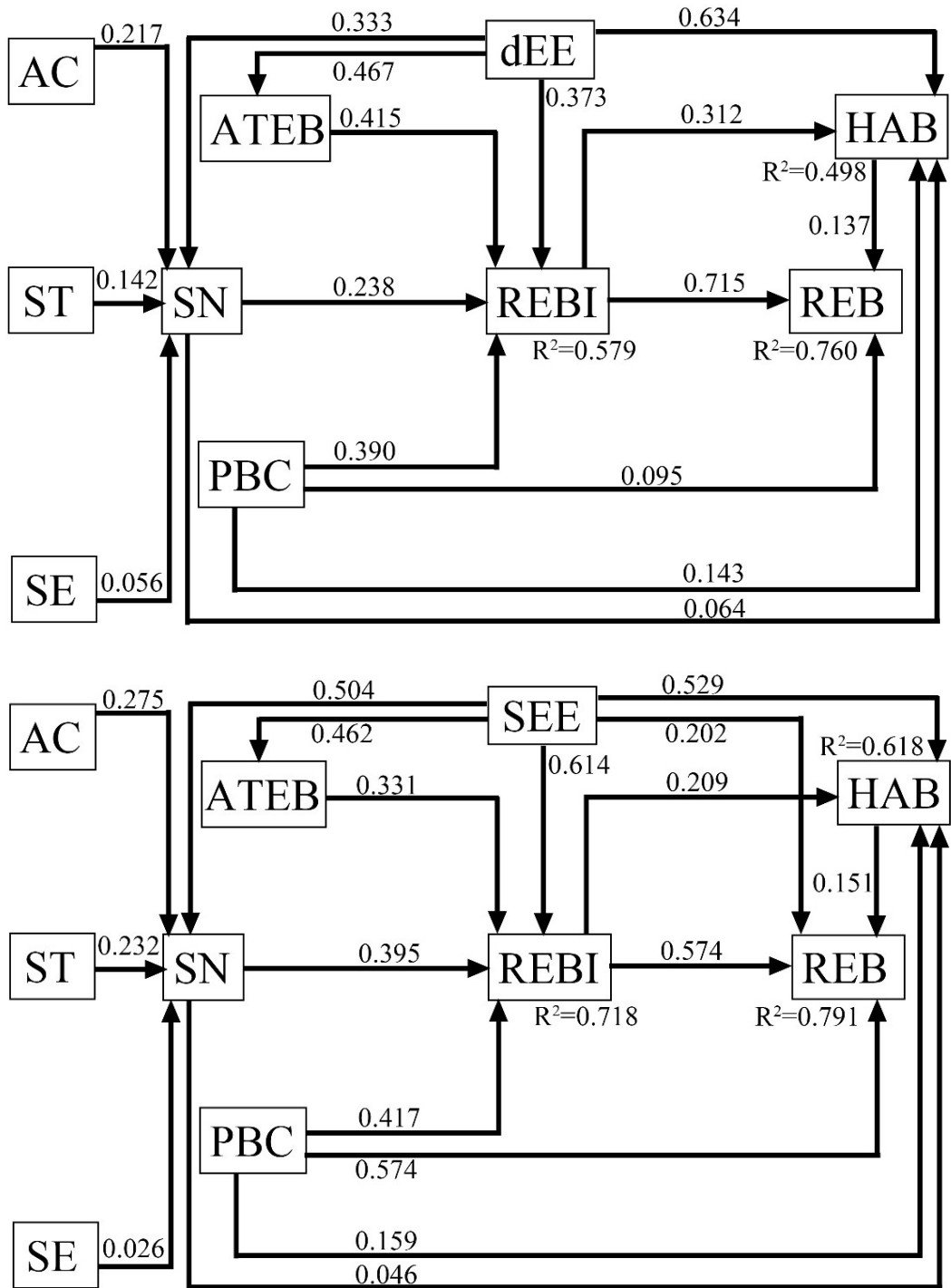

**Figure 2.** Results of the structural equation model.

The normalized path coefficient from dEE to REBI is 0.373; therefore, dEE and REBI are highly positively correlated. Hypothesis H10b is verified.

The normalized path coefficient from sEE to SN is 0.504, suggesting a high correlation between sEE and SN. Hypothesis H11a is verified.

The normalized path coefficient from dEE to SN is 0.333, indicating a high correlation between dEE and SN. Hypothesis H11b is verified.

The normalized path coefficient from sEE to ATEB is 0.462, indicating a high correlation between sEE and ATEB. Hypothesis H12a is verified.

The normalized path coefficient from dEE to ATEB is 0.467, suggesting a high correlation between dEE and ATEB. Hypothesis H12b is verified.

The normalized path coefficient from sEE to REB is 0.202, suggesting a high correlation between sEE and REB. Hypothesis H13a is verified.

The normalized path coefficient from sEE to HAB is 0.529, suggesting a high correlation between sEE and HAB. Hypothesis H14a is verified.

The normalized path coefficient from dEE to HAB is 0.634, suggesting a high correlation between dEE and HAB. Hypothesis H14b is verified.

## 5. Discussion

This research has made significant strides in understanding how environmental education can promote responsible environmental behavior among tourists, a key factor in driving sustainable development and travel. By revealing the pathways and differences between situational education and everyday environmental education, this study contributes to the formulation of sustainable development indicators and frameworks, as well as the broader objectives of promoting sustainable tourism.

### 5.1. Contribution to Sustainable Development Indicators and Frameworks

Quantifying the impact of environmental education: Our research results provide a detailed interpretation of how different types of environmental education can influence responsible environmental behavior. By quantifying these impacts, this study contributes to the development of measurable sustainability indicators that can assess the effectiveness of educational interventions. These indicators can be incorporated into broader sustainable development frameworks to evaluate the progress of the tourism industry in achieving environmental, social, and economic sustainability goals.

Refinement of a behavioral model: Integrating personal factors, such as attitudes, subjective norms, and perceived behavioral control, into our analysis enhances the complexity of the sustainable development framework. This improvement helps better predict and understand the mechanisms behind responsible environmental behavior, which are crucial for planning and implementing effective sustainable strategies in the tourism industry.

### 5.2. Suggestions for Promoting Sustainable Tourism

Our findings indicate that situational environmental education is more effective in promoting responsible environmental behavior through personal norms and intention, whereas daily environmental education primarily influences habits to encourage responsible environmental behavior. Moreover, a tourists' behavior is influenced by personal factors, such as attitudes and subjective norms, which are validated by existing studies [63,64], and the authors suggest some actions based on these findings.

1. Tourists' attitudes towards environmental behaviors are positively associated with their REBI. To promote responsible environmental behavior among tourists, scenic spots should prioritize maintaining cleanliness and aesthetic appeal, as this can encourage visitors to engage in pro-environmental actions. People-friendly services, including reasonable prices and standardized tours, should also be provided. Effective advertising campaigns can further raise awareness among tourists about the significance and value of scenic spots, thereby guiding their REB [65].
2. Tourists' SN is positively associated with their REBI. Thus, to leverage the positive impact of SN on REBI, relevant departments should proactively encourage schools, communities, and organizations to visit these locations, fostering a positive environmental atmosphere and inspiring tourists to engage in REB.
3. Tourists perceived behavioral control as a positive influence on both their REBI and REB. Therefore, to enhance tourists' PBC and promote responsible environmental behavior, the scenic area infrastructure should be comprehensive and well maintained.

Sufficient and user-friendly distribution of various signs and trash cans should be ensured throughout the area. In addition, installing warning signs in areas where uncivilized behavior is likely to occur can help improve tourists' PBC [66].

4.  The personal qualities of tourists, such as their awareness of consequences and superego values, have a positive impact on their SN. Therefore, the government should promote the values of environmental protection through laws, etc. to control environmentally unfriendly behaviors and actively encourage support for responsible environmental behaviors [67].

*5.3. Limitations and Future Research*

Generalizability: While this study offers valuable insights, the applicability of its findings across various cultural and geographical backgrounds requires further confirmation. Future research should consider a wider array of tourist populations and destinations to bolster the generalizability of the results.

Longitudinal Study: To truly comprehend the long-term impacts of environmental education on sustainable tourism, future research should employ a longitudinal design, tracking changes in tourist behavior over an extended time period after education.

Integration with Other Sustainable Development Strategies: Future research should also explore how environmental education can effectively be integrated with other sustainable development strategies, such as eco-friendly infrastructure development and policy changes, to more comprehensively promote sustainable tourism.

## 6. Conclusions

Environmental education is essential for solving environmental problems. In this study, we analyzed the pathways through which environmental education influences REB through a comprehensive model and identified the mechanisms by which daily environmental education and situational environmental education influence REB.

(a) Daily environmental education influences the pathways of tourists' responsible environmental behavior mainly through attitudes (0.467) and habits (0.634). Daily environmental education also has a positive influence on tourists' norms (0.333) as well as intentions (0.373).

(b) Situational environmental education influences the pathways of tourists' responsible environmental behavior mainly through habits (0.534), norms (0.504), and intention (0.614). There were also positive effects on behavior (0.202) and attitude (0.462).

(c) Personal tourist factors, such as ATEB, SN, and PBC, are strongly and positively associated with their REBI, which in turn significantly influences their REB.

(d) AC, ST, SE, and other factors have a positive effect on SN, but the effect is not significant.

In summary, this study highlights the key role of environmental education in promoting sustainable travel and enhancing sustainability indicators and frameworks. By gaining a clearer understanding of how different forms of education impact tourist behavior, we can better design and implement strategies, leading to more sustainable outcomes. As the tourism industry continues to evolve, the importance of integrating sustainability into every aspect of travel will become increasingly significant. We hope the findings of this research will contribute to creating a more sustainable future for the tourism industry and beyond.

**Author Contributions:** Conceptualization, J.W. and J.D.; data curation, J.W. and J.D.; investigation, J.W., X.Y., J.G., Y.W. and J.Z.; methodology, J.W.; software, J.W. and J.D.; supervision, B.J.D. and W.G.; visualization, J.D. and X.Y.; writing—original draft, J.W.; writing—review and editing, J.D.; funding, J.W. and X.Y. All authors have read and agreed to the published version of the manuscript.

**Funding:** The study was supported by the Kitakyushu Innovative Human Resource and Regional Development Program (JPMJSP2149). The study was supported by the 2023 Beilin District Science and Technology Plan Project—Applied Technology Research and Development (GX2339).

**Institutional Review Board Statement:** The study was conducted in accordance with the Declaration of Helsinki and it has been reviewed by the Faculty of Environmental Engineering, the University of Kitakyushu.

**Informed Consent Statement:** Informed consent was obtained from all subjects involved in the study.

**Data Availability Statement:** Data available on request due to privacy restrictions.

**Acknowledgments:** We appreciate the help of respondents and thank the Japan Science and Technology Agency project and the Science and Technology Bureau in Beilin District, Xi'an City, for supporting this study.

**Conflicts of Interest:** The authors declare no conflicts of interest.

## Abbreviations

| | |
|---|---|
| TPB | Theory of planned behavior |
| TRA | Theory of reasoned action |
| SN | Subjective norm |
| PBC | Perceptual behavior control |
| REB | Responsible environmental behavior |
| REBI | Responsible environmental behavioral intention |
| IBS | Inquiry-based science |
| ExE | Experiential education |
| OE | Outdoor education |
| GBL | Garden-based learning |
| ATEB | Attitude toward environmental behavior |
| HAB | Habit |
| ACs | Awareness of consequences |
| STs | Self-transcendence values |
| SEs | Self-enhancement values |
| dEE | Daily environmental education |
| sEE | Situational environment education |
| CFA | Confirmatory factor analysis |
| ESD | Education for sustainable development |
| EP | Environmental protection |
| CR | Composite reliability |
| AVE | Average variance |
| VR | Virtual reality |
| NAT | Norm Activation Theory |
| VBN | Value–Belief–Norm Theory |

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
