# Peer review of "Achieving Sustainable Tourism: Analysis of the Impact of Environmental Education on Tourists’ Responsible Behavior"

_sustainability, doi:10.3390/su16020552_

Round 1

Reviewer 1 Report

Comments and Suggestions for Authors

The peer-reviewed scientific study is undoubtedly a very interesting scientific work, the content of which can be beneficial not only for theory but also for practice.

The authors probably forgot to sufficiently read the instructions for authors available on the journal's website and strictly follow them so that the required content structure is observed.

The question is why are the keywords not sorted alphabetically and why are some written in capital letters?

The introduction lacks a clearly stated reason for writing this scientific study, a really clearly stated main goal and secondary goals. The hypotheses are established (their designation is very strange), but in a different part than the instructions for the authors indicate, and at the same time the answers to them are missing in the conclusion.

The mandatory part Theoretical basis/ review of the literature is missing.

  however, I consider the biggest shortcoming to be the fact that the topic of the manuscript does not correspond to the topic of the special issue, which is "Sustainability, Strategic Management, Smart Governance and Smart Cities"

this fundamental problem must be eliminated by modifying not only the name of the manuscript, but especially its content.

Reviewer 2 Report

Comments and Suggestions for Authors

This study, which is entitled "Context of Global Environmental Challenges: Influencing

Mechanisms and Processes of Environmental Education" discusses the challenges of environmental education. The study used quantitative analysis to examine the data collected from a tourist survey. The aim of the study, as stated in the abstract, was to investigate the impact of situational and daily environmental education on tourists, which is a major concern.

After reviewing the entire text of this study, it is not suitable for publication in its current form. I came to this conclusion because the manuscript needs to be better written scientifically. Instead, it is a report on data collected from a survey.

Second, the aim of this survey is unclear. Is the aim to promote environmental education in which school? Does it belong to the school of urban studies or landscape architecture? Or tourism? Which level of education would the author like tourists to receive? These questions leave the reader with unanswered questions.

Third, the discussion is very generic. All updated information should have been provided here. The limitations need to be better explained and guidance provided for future studies.

Fourth, what learning and teaching strategy was adopted in the tourism field to be investigated, and what previous studies implemented?

Reviewer 3 Report

Comments and Suggestions for Authors

The subject of the manuscript is within the scope of Sustainability, but it must undergo a drastic revision before being considered for publication.

To my knowledge the data given are original.

The title of the article is:

“Context of Global Environmental Challenges: Influencing Mechanisms and Processes of Environmental Education”

Since the article is focusing on responsible environmental behavior of tourists, to reflect the content the title should somewhat has “Tourist” word.

Although Travel Demand Management is among the keywords, the text does not cover anything related to this keyword. Authors should explain this issue. 

Under Introduction, it is stated that:

“Humanity is currently confronted with a range of critical environmental challenges that cannot be ignored, including climate change, depletion of natural resources, and biodiversity loss [1].….” “…..Researchers generally believe that current human behavior has a negative impact on the earth's environment [3]….”.

The references [1] and [3] are dated back to 2014. There are recent reports prepared after 2020. It is recommended to use these reports as references.

Introduction should be strengthened by quoting clearly what this study adds to the existing knowledge ie. clearly indicating the difference between this study and Wang, J., et al., Impact of Situational Environmental Education on Tourist Behavior—A Case Study of Water Culture Ecological Park in China. 2022. 19(18): p. 11388.

The following sentence requires references:

“…..In previous research, environmental education has often been treated as a covariate influencing attitudes…..”

In Table 1. The constructs and scale used in the study, item text is defined as “Preserving the tourist attraction environment is a happy thing for me.”. Does this sentence mean “Preserving the tourist attraction environment is something I find joy in.”?

The article is structured as

5. Results

6. Discussion and Conclusion

7. Limitations

It is recommended to have separate parts dedicated to

Results and Discussion (Limitations of the study can be discussed under this heading as well)

and

Conclusions

There are typo errors in the text, such as:

“….environmental issues. [26], the relationships….”

“….more as a situational factor Influence….”

These should be corrected.

Comments on the Quality of English Language

English should be improved. 

Round 2

Reviewer 1 Report

Comments and Suggestions for Authors

I am GLAG that  the authors fully  incorporatd  the comment of all  reviewrs.

Author Response

Dear Reviewer,

Thank you for your time and expertise in reviewing our manuscript. We deeply appreciate your contributions to enhancing the quality of our work.

Sincerely,

Jinming Wang and Jialu Dai

Reviewer 2 Report

Comments and Suggestions for Authors

In reviewing this revised version, I would like to acknowledge the authors' efforts to address the comments of all reviewers. Best

Author Response

Dear Reviewer,

Thank you for dedicating your time and expertise to review our manuscript. Your constructive feedback was invaluable in enhancing the quality of our work. 

Sincerely,

Jinming Wang and Jialu Dai

Reviewer 3 Report

Comments and Suggestions for Authors

The subject of the manuscript is within the scope of Sustainability. 

To my knowledge the data given are original.

The manuscript can be accepted in the present form.

Comments on the Quality of English Language

The authors improved the language. 

Author Response

Dear Reviewer,

Thank you for your positive feedback and for recognizing the improvements in our manuscript on Sustainability. We are pleased that you find the data original and the work suitable for acceptance. Your support is greatly appreciated.

Best regards,

Jinming Wang and Jialu Dai